# Gamma Irradiation Triggers Immune Escape in Glioma-Propagating Cells

**DOI:** 10.3390/cancers14112728

**Published:** 2022-05-31

**Authors:** Nicola Hoppmann, Nora Heinig, Ute Distler, Ella Kim, Volker Lennerz, Yvonne Krauß, Ulrike Schumann, Alf Giese, Stefan Tenzer, Lynn Bitar, Mirko H. H. Schmidt

**Affiliations:** 1Department of Neurology, Research Center Translational Neurosciences (FTN), Rhine-Main Neuroscience Network (rmn2), University Medical Center, Johannes Gutenberg University Mainz, Langenbeckstr. 1, 55131 Mainz, Germany; nicola.hoppmann@gmail.com; 2Institute of Anatomy, Medical Faculty Carl Gustav Carus, Technische Universität Dresden, Fetscherstr. 74, 01307 Dresden, Germany; nora.heinig@tu-dresden.de (N.H.); ulrike.schumann@tu-dresden.de (U.S.); 3Institute for Immunology, University Medical Center, Johannes Gutenberg University Mainz, Langenbeckstr. 1, 55131 Mainz, Germany; ute.distler@uni-mainz.de; 4Translational Neurooncology Research Group, Department of Neurosurgery, University Medical Center, Johannes Gutenberg University Mainz, Langenbeckstr. 1, 55131 Mainz, Germany; ella.kim@unimedizin-mainz.de (E.K.); alf.giese1@gmail.com (A.G.); 5Department of Medicine III, Hematology, Medical Oncology and Pneumology, University Medical Center, Johannes Gutenberg University Mainz, Obere Zahlbacher Str. 63, 55131 Mainz, Germany; lennerz@uni-mainz.de (V.L.); yeichinger@gmx.net (Y.K.)

**Keywords:** glioma-propagating cells, glioma stem cells, detergent-resistant membranes, radio-resistance, immune escape

## Abstract

**Simple Summary:**

Stem cell-like glioma-propagating cells (GPCs) are crucial for initiation, growth, and treatment resistance of glioblastoma multiforme. Due to their strong immunosuppressive activities, they essentially limit immunotherapeutic approaches. This study offers a new model of radio-selected patient-derived GPCs mimicking a clinical treatment regime of tumor irradiation which is especially useful for immunotherapeutic studies. We provide evidence that clinically relevant, sub-lethal fractions of γ radiation select for a more radio-resistant GPC phenotype with lower immunogenic potential, potentially hampering the success of adjuvant T-cell-based immunotherapies. The immune evasion in GPCs was characterized by quantitative proteomics. It revealed a marked downregulation of the antigen processing machinery in lipid rafts of these cells, leading to reduced MHC surface expression and weaker cytotoxic T lymphocyte (CTL) recognition.

**Abstract:**

Glioblastoma multiforme is the most common and devastating form of brain tumor for which only palliative radio- and chemotherapy exists. Although some clinical studies on vaccination approaches have shown promising efficacy due to their potential to generate long-term immune surveillance against cancer cells, the evasion mechanisms preventing therapy response are largely uncharacterized. Here, we studied the response of glioblastoma-propagating cells (GPCs) to clinically relevant doses of γ radiation. GPCs were treated with 2.5 Gy of γ radiation in seven consecutive cellular passages to select for GPCs with increased colony-forming properties and intrinsic or radiation-induced resistance (rsGPCs). Quantitative proteomic analysis of the cellular signaling platforms of the detergent-resistant membranes (lipid rafts) in GPCs vs. rsGPCs revealed a downregulation of the MHC class I antigen-processing and -presentation machinery. Importantly, the radio-selected GPCs showed reduced susceptibility towards cytotoxic CD8+ T-cell-mediated killing. While previous studies suggested that high-dose irradiation results in enhanced antigen presentation, we demonstrated that clinically relevant sub-lethal fractionated irradiation results in reduced expression of components of the MHC class I antigen-processing and -presentation pathway leading to immune escape.

## 1. Introduction

Glioblastoma multiforme (GBM) is the most common and most malignant astrocytic tumor of the central nervous system (CNS) in adults. The multimodal treatments, including surgery and combined radio- (RT) and temozolomide (TMZ) chemotherapy, are only palliative and prolong patients’ median survival from 12.1 months after radiotherapy alone to 14.6 months [1]. In addition to cellular heterogeneity, invasiveness, and genetic instability, therapy failures are attributed to stem cell-like glioma-propagating cells (GPCs). These cells show resistance to irradiation as well as chemotherapy and have been suggested to be responsible for the recurrence of more aggressive tumors after therapy [2,3,4,5]. T-cell-based immunotherapies have emerged as promising therapeutic strategies as they exploit the immune system’s ability to specifically recognize and eliminate malignant cells. However, their therapeutic efficacy is limited due to strong immunosuppressive activities of the GPCs and the generation of a specialized protective microenvironment [6,7,8,9,10]. A combination of gamma γ-irradiation and immunotherapy has been proposed as a treatment, because single high-dose irradiation in vitro increases the expression of major histocompatibility complex (MHC) class I molecules on tumor cell surfaces, improving their recognition by cytotoxic CD8+ T cells [11,12,13]. However, this single-dose irradiation protocol, which includes monitoring of irradiation-induced effects 2–6 days after treatment, does not efficiently mimic clinical fractionated irradiation, and thus does not address long-term effects on tumor phenotype. Fractionation of high-dose γ-irradiation is clinically required to reduce side effects. To consider this necessity, individual irradiations are performed at sub-lethal doses that have been suggested to select for GPCs with intrinsic or acquired resistance to further treatment [14,15,16]. Additionally, therapeutic timing must be considered, as combinatorial radiotherapy with TMZ chemotherapy leads to lymphopenia, possibly hampering the success of additional immunotherapy [17]. Additionally, biophysical preclinical models predicting the optimum timing of irradiation combined with the administration of checkpoint inhibitors to achieve a robust immune response proved to be efficient [18].

Known effector mechanisms of ionizing radiation include apoptosis induction by direct DNA damage but also by impacting membrane integrity and the composition of membrane signaling platforms, i.e., lipid rafts (LR) or detergent-resistant membranes (DRM) [4,19,20]. The exact molecular pathways resulting in the acquisition of a resistant phenotype in irradiated GPCs are largely unknown. However, as GPCs are presumably responsible for tumor recurrence and therapy resistance, targeting this subpopulation will be critical for successful glioma therapy [21,22].

In this study, we describe a novel model of patient-derived GPCs, which we subjected to a clinically relevant fractionated radiation scheme (2.5 Gy in seven consecutive cellular passages) to generate a derivative GPC line (rsGPC) that displayed increased radiation resistance and higher colony-forming potential. These selected radioresistant cells mimic recurrent cells due to the fact that after radiotherapy most of these recurrences originate from the marginal zone of the irradiated field [23]. Using a label-free quantitative proteomics approach on isolated DRMs from the parent and radio-selected GPC lines, we observed a marked downregulation of proteins involved in antigen processing and presentation, which resulted in a decreased expression of MHC class I molecules on the cell surfaces of radio-selected GPCs. This concomitantly reduced susceptibility of radio-selected GPCs to lysis by cytotoxic CD8+ T cells. Taken together, our data show that the application of fractionated radiation preferentially selects GPCs with a phenotype characterized by an enhanced resistance to radiation and diminished immunogenic potential.

## 2. Materials and Methods

### 2.1. Culture and Irradiation of Human GPCs

Human glioma sphere cultures were established from glioblastoma specimens obtained at the University Medical Center Göttingen (UMG) in accordance with the UMG ethical review board as described previously [24]. Briefly, tumor tissue was dissociated and maintained in Neurobasal-A medium (NB) supplemented with B27 supplement (Invitrogen), fibroblast growth factor-2 (10 ng/mL, R&D), and epidermal growth factor (20 ng/mL, R&D) in 0.1% BSA (NB complete). The tumorigenic potential was tested in an orthotopic xenograft mouse model for GBM. In brief, single-cell suspensions were prepared from gliomasphere cultures by using a combined trypsin/mechanical trituration procedure. Cells were washed twice in PBS and re-suspended in PBS at 2 × 10^4^ cells/μL. Cell viability was determined by trypan blue staining. A volume of 5 µL of a single-cell suspension (cell vitality >95%) were injected into the caudato-putamen of the right hemisphere using the following stereotactic coordinates in reference to the bregma: 1 mm (anteroposterior axis), 3 mm (lateromedial axis), 2.5 mm (vertical axis). Mice were sacrificed at the first manifestation of neurological symptoms. Tumor-bearing mouse brains were extracted and fixed in 4% paraformaldehyde in PBS for at least 24 h at +4 °C. After fixation, brains were paraffin embedded, dissected into 1–3 μm thick coronal sections, and analyzed by immunohistochemical staining using antibodies specific to human nestin (R&D Systems GmbH, Wiesbaden-Nordenstadt, Germany).

Radioresistant sub-lines were isolated by subjecting single-cell suspensions of GPCs to 2.5 Gy of ionizing radiation at 1 Gy/min repeated during seven consecutive rounds of passaging. All experiments were performed 13–33 passages after the final irradiation.

### 2.2. Colony Formation Assay

Overall cell survival of rsGPCs and control GPCs in response to indicated doses of γ radiation was measured by colony formation. The spheroids were thoroughly dissociated with Accutase to prepare single-cell suspensions. Identical numbers of cells were left untreated or irradiated with 2.5, 5.0, or 7.5 Gy of ionizing radiation. The 500 cells/well of a 24-well plate were plated with 2 mL of NB complete medium. After 2 weeks, the number of colonies with more than 50 cells per colony was counted. The experiment was performed three times with 4 replicates.

### 2.3. PBMC and CD8+ T-Cell Isolation

Patient-specific tumor HLA molecules were determined (HLA-A*01:01, A*02:01; B*08:01, B*15:01; C*03:03, C*07:01) in a HLA diagnostics laboratory (Dr. Thiele, Institute for Immunology and Genetics, Kaiserslautern, Germany). A blood donation from a HLA class I-partly matched healthy donor (HLA-A*01:01, A*02:01, B*08:01, B*15:18, C*07:01, C*07:04) was obtained from the German Red Cross Blood Donation Center (Bad Kreuznach, Germany). The blood was diluted with an equal amount of PBS without Ca^2+^/Mg^2+^ and 25 mL of blood-PBS mixture were carefully layered on top of 15 mL of Lymphoprep^TM^ and centrifuged at RT and 764× *g* for 40 min without active braking. The lymphocyte rings (intermediate phase) were transferred into new 50 mL tubes, washed with 40 mL of PBS, and centrifuged at RT and 561× *g* for 15 min. Supernatant was discarded and the pellet resuspended in 10 mL of RPMI wash medium (RPMI 1640 supplemented with 5% FCS, 1% HEPES, Pen/Strep) for cell counting. A total of 1 × 10^8^ peripheral blood mononuclear cells (PBMCs) were used for subsequent CD8+ T-cell isolation using the Miltenyi MACSBeads CD8+ T cell Isolation Kit (Bergisch Gladbach, Germany) according to the manufacturer’s protocol.

### 2.4. Mixed Lymphocyte Tumor Cultures

Mixed lymphocyte tumor cultures (MLTCs) were established to generate GPC-reactive human CD8+ T cells. Spheroids of GPCs and rsGPCs, serving as APCs/target cells, were singularized and incubated with 1 µg/mL of IFNγ 24 h prior to experiment to increase HLA expression. The next day, tumor cells were resuspended in 10 mL of human AB medium (RPMI 1640 supplemented with 5% human AB serum, 1% HEPES, L-Glutamine, Pen/Strep) and were irradiated with 120 Gy. For the first incubation week, autologous CD8-negative T cells were used as feeder cells, which were resuspended in 10 mL of AB medium and treated with 30 Gy gamma radiation. The 10^4^ tumor cells, feeder cells and cytolytic CD8+ T cells were plated in a U-shaped 96-well plate in 200 µL of AB medium per well, supplemented with IL-7 (5 ng/mL), IL-12 (1 ng/mL) and IL-15 (5 ng/mL). CD8+ T cells were stimulated weekly with freshly irradiated and IFNγ-stimulated GPCs as well as IL-7 and IL-15 by maintaining T cell to tumor cell ratio of 1:1–1:5. IL-12 was replaced by IL-2 (100 IU/mL) from the 2nd restimulation onwards.

If necessary, half of the medium was replaced by fresh medium between two stimulations without addition of cytokines. On culture day 19, all microcultures were tested for tumor reactivity in an IFNγELISPOT assay to select for tumor-reactive cultures. Ten microcultures per tumor cell line (GPC and rsGPC) with high IFNγ-producing cells were continued to be stimulated for further expansion and were tested for tumor reactivity in a ^51^chromium (^51^Cr) release assay on culture day 40.

### 2.5. Flow Cytometry

Surface expression was determined by flow cytometric analysis. A total of 100,000–200,000 cells per sample were stained with primary and secondary antibodies at 4 °C for 20 min. Primary antibodies used were monoclonal antibodies PA2.1 (anti-HLA-A2 [25]) B1.23.2 (anti-HLA-B/C [26]), antibodies against HLA-A1 and HLA-B15 (both purchased from One Lambda, West Hills, Los Angeles, CA, USA), as well as those from the IOTest Beta Mark TCR Vbeta Repertoire Kit (Beckman Coulter, Krefeld, Germany). The secondary antibody used was goat anti-mouse IgG conjugated with FITC (Beckman Coulter). Cells were fixed with 2% PFA at 4 °C for 20 min and HLA expression was measured by flow cytometry on a FACSCanto II device (BD Bioscience, Heidelberg, Germany). Data were analyzed using the FlowJo Software (Tree Star Inc., Ashland, OR, USA).

### 2.6. Isolation of Detergent-Resistant Membrane Fractions

A total of 1 × 10^8^ GPCs or rsGPCs were lysed in 0.5% Brij98^®^ lysis buffer for 45 min on ice. Lysis was accelerated by slowly pipetting up and down. Sucrose was added to each lysate for a final concentration of 50% and carefully solubilized by horizontal rotation. A discontinuous sucrose gradient was prepared in 12 mL ultracentrifuge tubes (Beckman Coulter) with 1 mL 50%, 5 mL 30%, and 5 mL 5% sucrose. Floating ultracentrifugation was carried out at 4 °C and 40,000 rpm for 18 h without braking in an ultracentrifuge (Optima L-80 XP, Beckman Coulter) with a SW-40 Ti swing rotor (Beckman Coulter). DRM fractions were located at the interface of 30% and 5% sucrose. Then, 500 µL fractions were harvested from top to bottom of the gradient and were analyzed by Western blot.

### 2.7. SDS-PAGE and Western Blot

The protein concentration of each fraction was measured by a Bradford protein assay. An amount of 5 µg protein of each fraction was separated on 10% polyacrylamide gels at 40 mA per gel for 2 h and were blotted onto nitrocellulose membranes by semi-dry Western blot at 180 mA per membrane for 1 h. Membranes were blocked in 5% skim milk powder in TBS-T, incubated with the indicated primary antibody solution (in TBS-T supplemented with 5% BSA and 0.1% sodium azide) at 4 °C overnight followed by incubation with corresponding HRP-conjugated secondary antibody solution (in 3% BSA in TBS-T) at RT for 1 h. Proteins were visualized with ECL Detection Reagents (GE Healthcare, Chicago, IL, USA). The antibodies used were anti-beta Actin (MP, Irvine, CA, USA), anti-Tapasin, anti-beta2-Microglobulin, anti-ERp57, anti-Calreticulin (all Abcam, Cambridge, UK), anti-Yes (BD Bioscience), anti-Flotillin1, anti-Caveolin1 (Sigma-Aldrich, Seelze, Germany), anti-TAP1 (Novus Biologicals, Cambridge, UK), anti-TAP2 (MBL, Woburn, MA, USA), HRP-conjugated goat anti mouse and HRP-conjugated goat anti-rabbit (Thermo Scientific, Bonn, Germany).

### 2.8. Mass Spectrometric Analysis

Aliquots of isolated membrane fractions corresponding to 20 µg protein were digested as described previously [27]. Prior to LC–MS analysis, the resulting tryptic digest solutions were diluted to a concentration of 500 ng/µL using aqueous 0.1% *v*/*v* formic acid and spiked with 25 fmol/µL of enolase 1 (*Saccharomyces cerevisiae*, Waters GmbH, Eschborn, Germany) tryptic digest standard. Nano LC–MS analysis of tryptic peptides was performed using a nanoAcquity UPLC system (Waters) coupled to a Waters Q-TOF Premier API system (Waters). Peptides were separated using a 75 µm × 150 mm BEH-C18 reversed phase column. Mobile phase A was water containing 0.1% formic acid and mobile phase B was acetonitrile containing 0.1% formic acid. Samples (2.6 µL per injection) were loaded onto the column in the direct injection mode with 1% mobile phase B as described before [28]. Peptides were separated using a gradient from 1% to 35% mobile phase B over 110 min at a flow rate of 300 nL/min. After separation of peptides, the column was rinsed with 90% mobile phase B, followed by a re-equilibration step at initial conditions (1% mobile phase B), resulting in a total run time of 150 min. [Glu1]-fibrinopeptide was used as lock mass at 500 fmol/µL. Samples were analyzed in three technical replicates. Nano ESI–MS analysis of tryptic peptides on the Waters Q-TOF Premier API system was performed in the positive V-mode with a resolving power of at least 10,000. The instrument was equipped with a NanoLockSpray source and the lock mass channel was sampled every 30 s. For fragment identification and relative quantification of the peptide fragments, the instrument was run in the elevated energy (MS^E^) acquisition mode [29]. In the low-energy MS mode, data were collected at constant collision energy of 3 eV. Collision energy was ramped from 16 to 32 eV in the MS^E^ scan. The spectral acquisition time in each mode was 0.7 s with a 0.05 s interscan delay. One cycle of MS and MS^E^ data was acquired every 1.5 s. Continuous raw data processing was performed using ProteinLynx Global Server (PLGS) version 2.5.2 (Waters Corporation, Milford, MA, USA). Peptide and protein identifications were obtained by searching a database containing UniprotKB/Swissprot entries of the human reference proteome (UniProtKB release 2011_08, 20,244 entries). Sequence information of enolase 1 (*S. cerevisiae*) and porcine trypsin was added to the database. The experimental data were searched in PLGS2.5.2, using precursor and fragment ion mass tolerances automatically determined by PLGS2.5.2 during the database search. Further on, the following parameters were used for the database search: trypsin was set as the digestion enzyme with one missed cleavage allowed and fixed carbamidomethylcysteine and variable methionine oxidation were set as the modifications. The false discovery rate (FDR) for peptide and protein identification was assessed by searching a randomized database, which was automatically generated by PLGS 2.5.2 by randomizing the sequence of each entry. The FDR was set to 1% threshold for database search in PLGS.

Data were post-processed using the in-house-developed software ISOQuant as described previously in detail [30]. Post-identification analysis included retention time alignment, exact mass and retention time (EMRT) clustering. Only proteins identified by at least two peptides with a minimum length of six amino acids were considered.

### 2.9. Data and Statistical Analysis

Enrichment of distinct signaling cascades was visualized using KEGG signaling pathway (http://www.genome.jp/kegg/; accessed on 1 January 2014) and DAVID functional annotation tool (http://david.abcc.ncifcrf.gov/; accessed on 1 January 2014) [31]. All data were analyzed using PRISM5 (Graphpad software). Data are presented as the mean ± SD. Statistical analysis of the data was conducted using a parametric *t*-test with Welch’s correction. Results with a *p*-value < 0.05 were considered significant. * *p* < 0.05; ** *p* < 0.01; *** *p* < 0.001; **** *p* < 0.0001.

## 3. Results

### 3.1. Radio-Selection and Phenotypic Analysis of GPCs

To investigate the effect of clinically relevant fractionated radiation on GPCs, we selected for GPCs with intrinsic or radiation-induced resistance (radio-selected GPCs, rsGPC) by treating the cells with 2.5 Gy of γ radiation in seven consecutive cellular passages while control cells were left untreated (shown schematically in Figure 1A). The application of different daily doses allowed an adaptation of the radiation intensity to the cell line by analyzing the capacity of GPC cells for spheroid formation. After 7 × 2.5 Gy radiation, 44% (44% ± 2.66, *n* = 3) of colonies were formed compared to untreated GPCs (Appendix A). A drastic reduction in colony formation capacity was observed for an application of 7 × 5 Gy (9% ± 1.96, *n* = 3), and almost no colonies were detected for 7 × 7.5 and 7 × 10 Gy radiation (2% ± 0.52 and 0.3% ± 0.11, *n* = 3). Lower doses between 0 and 2.5 Gy were not analyzed due to the lack of comparability with standard radiotherapy. We subsequently explored the impact of radio-selection on colony formation capacities, resulting in significantly increased colony formation capacities of rsGPCs in contrast to the untreated control GPCs (Figure 1B). To test whether there is a correspondence between the capacity to form colonies in vitro and the tumorigenic properties, we next compared tumor growth rates of untreated GPCs and rsGPCs in the orthotopic model for glioma. The tumor growth potential was determined as the average survival time for development of neurologically symptomatic tumors. The results showed that xenografts derived from rsGPCs grew considerably faster (56 ± 8.83 days, *n* = 6) than those generated by untreated GPCs (90 ± 21.32 days, *n* = 4). Tumors derived from either GPCs or rsGPCs express the neural stem cell marker nestin and manifest intratumoral histomorphological heterogeneity characteristic of GBMs (Figure 1C). Next, we challenged both groups with increasing doses of γ radiation. Increasing doses of γ radiation reduced the number of colonies formed of both, rsGPCs and control GPCs. Nevertheless, rsGPCs showed significantly higher (*p*-value * < 0.05) colony formation capacities at irradiation doses of 2.5, 5.0, and 7.5 Gy as compared to control GPCs (Figure 1D). The results suggest that the fractionated radiation selected for GPCs with increased colony formation potential or prompted an increased colony formation potential.

### 3.2. Comparative Analysis of DRM Composition

Previous studies have shown that γ radiation affects the composition of lipid rafts and downstream signaling, e.g., defects in the formation of apoptosis-inducing ceramides have been shown to confer radio-resistance to lymphoblasts from Niemann–Pick patients and to a human head and neck squamous carcinoma cell line [19,20] To investigate quantitative changes in the composition of DRMs induced by radio-selection, we isolated DRMs from GPCs and rsGPCs. First, cellular compartments were separated by ultracentrifugation on a floating discontinuous sucrose gradient. Subsequently, fractions were analyzed by Western blot to identify fractions containing DRM-associated proteins (Figure 2A). Fractions containing the lipid raft markers Yes, Flotilin 1 (Flot1), and Caveolin 1 (Cav1) were subjected to a label-free quantitative proteomic analysis. In total, 454 proteins could be identified and quantified in the isolated DRM fractions. Interestingly, the overall composition of DRM-associated proteins was largely unaffected by the fractionated irradiation scheme. However, our analysis revealed an irradiation-induced more than two-fold reduction in individual proteins, including the zinc transporter ZnT7, the MHC class I molecule HLA-B8, Tapasin, and the H+/K+ ATPase ATP12A (Appendix A). This illustrates the downregulation of cell surface molecules involved in antigen presentation post-irradiation.

### 3.3. Downregulation of MHC Class I Antigen-Processing and -Presentation Components in rsGPC

A detailed evaluation of the identified DRM-associated proteins by Kyoto Encyclopedia of Genes and Genomes (KEGG) pathway analysis revealed a prominent downregulation of proteins related to the major histocompatibility complex (MHC) class I peptide loading and antigen-presentation machinery in rsGPCs as compared to GPCs (Figure 2B,C). We verified the decreased expression of antigen-processing components observed in our quantitative proteomics approach using Western Blot (Figure 2D) and qRT-PCR analysis (Appendix A) of GPCs and rsGPCs. Notably, our analyses confirmed a significantly decreased expression of both monomers of the heterodimeric channel protein antigen peptide transporter (TAP1, TAP2) in rsGPCs on the protein level, which could also be observed on the mRNA level. Additionally, we confirmed decreased expression of the endoplasmic reticulum resident protein 57 (ERp57), Tapasin (Tpn), and β2-Microglobulin (β2-M) but not of Calreticulin (Figure 2D). In summary, our data indicate that fractionated radiation diminished expression of components of the MHC class I antigen-processing and -presentation machinery in rsGPCs.

### 3.4. Evaluation of MHC Class I Surface Expression and Recognition Potential by Cytotoxic CD8+ T Cells after Fractionated Radiation

The observed downregulation of the antigen-processing and -presentation machinery in rsGPCs prompted us to investigate the impact on MHC class I expression on the cellular surface. After the determination of patient-specific human leukocyte antigen (HLA) alleles, we analyzed the expression of HLA class I molecules on the surface of GPCs by flow cytometry using HLA-subgroup- and single-allele-specific monoclonal antibodies. Our analysis revealed a significantly decreased expression of HLA-A2, HLA-B15, and HLA-BC alleles on rsGPCs in contrast to untreated control GPCs (Figure 3A,B), suggesting a lower immunogenic potential of radio-selected GPCs.

To investigate whether the decreased HLA class I expression on the surface of rsGPCs diminished the recognition potential of these cells by cytotoxic CD8+ T lymphocytes (CTL), we established MLTCs from a HLA class I-partly matched healthy donor. Tumor-reactive T-cell cultures were selected using an IFNγELISPOT assay. Cultures were established in a 96-well microculture format to favor stimulation of rare peptide-dependent minor histocompatibility antigens (mHAg)- or tumor antigen-specific CTLs instead of allo-HLA-B or -HLA-C mismatch reactions, which would dominate in bulk MLTC. After six rounds (equal to 40 d) of stimulation with GPCs or rsGPCs, the CTLs were tested for their ability to lyse GPCs and rsGPCs in a ^51^Cr release assay. The lysis potential was assessed using various effector-to-target ratios (ranging from 20:1 to 1:1). Interestingly, CTL cultures exhibited a significantly (*p*-value range: <0.001–<0.05) higher lytic activity against control GPCs as compared to rsGPCs (Figure 3C; Appendix A), independent of the tumor cells with which they had been stimulated. However, the effects were most prominent for cultures raised against rsGPCs. Along the same lines, we observed that two of these cultures (C1 and C2) were not able to lyse rsGPCs but control GPCs. These results further confirmed that rsGPCs displayed reduced immunogenicity.

To clarify whether the CTLs recognized the same antigen on GPCs and rsGPCs, we exemplarily analyzed the clonality of one of the CTL microcultures raised against rsGPCs (C4) by flow cytometry with a panel of antibodies against 24 T-cell receptor variable beta (Vβ) chains (Appendix A). Our results indicate that the C4 CTL culture was highly enriched for TCR-Vβ17-positive T cells (78.3%), while potential contaminations by natural killer (NK) cells (CD3−, CD16+, and CD56+) could not be detected. The enrichment of a specific TCR beta chain and the fact that lytic ability above 20% was already observed at a low effector-to-target ratio of 2:1 suggest that this CTL population was responsible for the tumor cell lysis.

In summary, all tested CTL cultures showed higher cytotoxicity towards control GPCs as compared to rsGPCs. This indicates that the reduced expression of HLA molecules induced by fractionated radiation led to diminished recognition of rsGPCs by immune cells.

## 4. Discussion

Despite intensive research and aggressive multimodal treatments, glioblastoma multiforme remains a lethal brain tumor. Stem cell-like GPCs are regarded as one origin in glioblastoma multiforme and are the main players in tumor relapse due to their high resistance to radio-chemotherapy [32,33]. Combinations of standard treatment with dendritic cell vaccinations are currently being tested in clinical trials and some patients have shown promising systemic antigen-specific cytotoxicity and intratumor infiltration of cytotoxic CD8+ T cells. However, this is not always correlated with clinical improvement because GBMs display multiple immune suppression and evasion mechanisms [34]. In this study, we analyzed the effects of clinically relevant doses of fractionated radiation on GPCs. For this purpose, we investigated a model of patient-derived GPCs that were treated with clinically relevant doses of fractionated radiation (2.5 Gy) in seven consecutive cellular passages to select for GPCs with higher intrinsic or radiation-induced resistance mechanisms. The irradiation dose was adapted according to clinical relevance, the spheroide formation capacity, and usual doses for in vitro fractionated radiation [23,35]. In standard radiotherapy of newly diagnosed glioblastoma, 1.8–2 Gy/fraction with a total dose of 50 to 60 Gy is applied according to the Stupp protocol [1]. However, particularly for older people with poor prognosis, a lower total irradiation dose with higher fractions (15 × 2.67 Gy) is used [36]. Furthermore, there is evidence that lower doses of irradiation increase the response to systemic agents [37]. That implies that the sufficient therapeutic dose of radiation depends among other things on different restrictions, the health status of the patients, and the combination with additional therapeutics. A comparative label-free quantitative proteomic analysis of lipid raft fractions showed that fractionated radiation resulted in an altered lipid raft composition and a marked downregulation of proteins involved in MHC class I antigen processing and presentation. This led to a significantly lower MHC class I surface expression and concomitantly resulted in a lower immune recognition potential of GPCs by cytotoxic CD8 + T cells.

Cancer propagating/progenitor cells have been suggested to resist conventional radiotherapy due to high free-radical scavenger levels [38], activation of the WNT/β-catenin signaling pathway [39], effective DNA damage repair mechanisms [2], and induction of autophagy [40]. Our data indicate that sub-lethal fractionated radiation can select for GPCs with intrinsic or acquired radio-resistance as well as GPCs with enhanced colony-formation capacities. Thus, the selection of more aggressive tumor cells due to sub-lethal treatment doses might hamper the efficacy of radiotherapy in recurrent GBM tumors. Our results are supported by several studies demonstrating that sub-lethal irradiation doses can select for a more resistant phenotype in GPCs with increased migratory and proliferation potential and resistance to apoptosis by activation of c-MET and NOTCH signaling [14,16,41].

Ionizing radiation does not only induce apoptosis by induction of DNA double-strand breaks, but also by induction of ceramide formation in the plasma membrane [42]. Our analysis of DRMs showed a reduced abundance of the zinc transporter ZnT7 (encoded by *Slc30a7*) as well as the potassium channel H+/K+ ATPase ATP12A within DRMs isolated from rsGPCs as compared to GPCs. ZnT7 is located in the Golgi apparatus membrane and is responsible for the transport of zinc from the cytosol into the Golgi complex [43]. Interestingly, the *Znt7*-knockout increased tumorigenesis in a transgenic mouse model of prostate adenocarcinoma (TRAMP/Znt7^−/−^), indicating that a reduced expression of this gene might contribute to tumor progression. Similarly, potassium channels have been linked to tumorigenesis [44] and increased expression of ATP12A (also known as ATP1AL1) has been associated with colorectal carcinomas [45]. Of note, recent findings suggest an association in the expression pattern of ATP12A changing from membrane bound to cytosolic within the prostate tumor in contrast to healthy prostate tissue, which shows no overall increase in expression [46]. While more detailed analyses are needed, changes in the intracellular signaling platforms might contribute to tumor progression and radio-resistance.

In this study, we focused on the evaluation of immune-relevant long-term effects of fractionated ionizing radiation. Our data showed a markedly reduced expression of proteins involved in MHC class I antigen processing and presentation including TAP1, TAP2, β2-Microglobulin, ERp57, and Tapasin. Notably, downregulation of TAP1 and TAP2 has been described as an immune evasion mechanism in gliomas [47], a subtype of breast cancer [48], and murine fibroblasts after oncogenic transformation [49], similar to the downregulation of β2-Microglobulin in diffuse large B cell lymphomas and colorectal carcinomas [50,51].

Of note, the immune-modulating properties of ionizing radiation might have opposed immunological effects for different radiation intensities. This is a rationale for the treatment of cancer by a combination of radiotherapy and immunotherapy. Ultra-fractionated radiation (below 1 Gy/fraction) has proven to be immunogenic whereas intermediate sub-lethal dose (2.5 Gy/fraction) enables glioma-propagating cells to become more resistant to radiotherapy with lower immunogenic potential [52]. Low-dose radiotherapy promotes the accumulation of macrophages and predominantly skews them toward a pro-inflammatory M1-like phenotype in the tumor microenvironment (TME) [52]. They normalize dysfunctional vessels enhancing a pro-inflammatory milieu [52]. An intermediate dose attenuates the pro-inflammatory phenotype and has minimal effect on cell survival. This can be explained by the fact that low irradiation doses are not severe enough to affect the viability and phagocytic functionality of activated macrophages which are necessary elements in the TME [53]. In addition, a moderate dosage did not affect the expression of pro- and anti-inflammatory markers, meaning, it was incapable of reprogramming tumor-associated macrophages [54]. This suggests that the combination with T-cell-based immunotherapy is not promising—especially concerning tumor-propagating cells and thus the recurrence potential. Understanding the impact of radiation doses on immune activation is a key for therapeutic efficacy but needs further clarification.

Our data demonstrate for the first time that fractionated radiation has a profound effect on the antigen-processing machinery in GBM. This reduced surface expression of MHC class I molecules led to significantly diminished recognition of rsGPCs as compared to control GPCs by partly HLA class I-matched CTLs. As our flow cytometric analysis on TCR variable beta chains of our exemplarily analyzed C4 CTL culture showed strong enrichment of TCR-Vβ17-positive T cells (78.3%), it is likely that the same MHC class I-peptide complexes were recognized on rsGPCs and control GPCs at least by C4 responder lymphocytes, also because the high lysis potential (>20% lysis) at an effector-to-target ratio as low as 2:1 suggested that this T-cell population was responsible for the specific lysis of rsGPCs and control GPCs. The identification of the individual, potentially GBM-specific MHC-peptide ligands will be a matter of future investigation, which is clearly hampered by the low proliferation rate of GPCs.

Taken together, our data indicate that clinically relevant doses of fractionated radiation triggered an immune escape mechanism in GPCs. A reduction in MHC class I molecule expression after γ radiation has been described in Ewing’s sarcoma cells due to reciprocal activation of amyloid precursor-like protein 2 (APLP2) [55]. However, our DRM analysis did not reveal an upregulation of APLP2 in rsGPCs as compared to control GPCs, but a direct downregulation of the antigen-processing machinery. Previous studies proposed a combination of radiotherapy with immunotherapy as single-dose γ radiation between 1–100 Gy enhanced protein degradation, leading to a larger peptide pool available for antigen presentation, increased MHC class I-restricted peptide presentation on the cellular surface, and better recognition by cytolytic CD8+ T cells [11,12]. However, while single doses of ionizing radiation likely improve immune recognition in the short term, they do not efficiently mimic a clinical treatment regime. Additionally, our study for the first time investigates the persisting effects of fractionated irradiation on GPCs, which could be essential in a therapeutic setting in which patients need to recover from radio- and chemotherapy before receiving dendritic cell vaccinations to boost anti-tumor immunity [56]. Induction of lymphopenia by radio-chemotherapy has been shown to be long lasting and has been associated with shorter overall survival in glioma [17,57] and squamous head and neck cancer [58].

Therefore, our new GBM model provides significant advantages over models based on single-irradiation protocols with a short-term readout. The latter may not be optimally suited for assessing long-term immunological aspects, as pretreatment lymphopenia has been documented to be a poor prognostic factor in patients with carcinomas, sarcomas, and lymphomas [59].

## 5. Conclusions

In summary, we characterized a novel immune evasion mechanism in a new model of glioma-propagating cells and radio-selected counterparts. We provide evidence that clinically relevant, sub-lethal fractions of γ radiation select for a more radio-resistant GPC phenotype with lower immunogenic potential, potentially hampering the success of adjuvant T-cell-based immunotherapies.

## Figures and Tables

**Figure 1 cancers-14-02728-f001:**
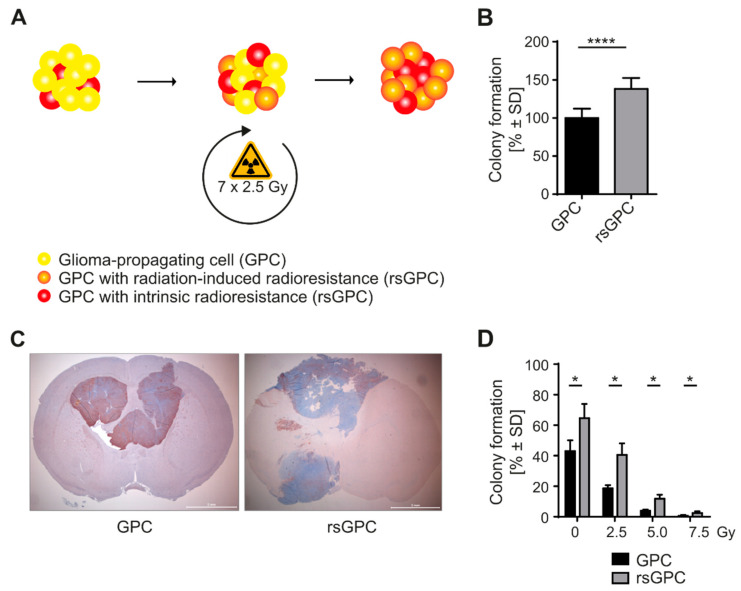
Generation of rsGPCs and challenge towards γ radiation. (**A**) Tumor tissue of GBM patients was dissected and single cells cultured in medium under neural stem cell conditions. GPC spheroids were separated and irradiated with 2.5 Gy (1 Gy/min) in 7 consecutive passages to select for a phenotype with intrinsic or acquired radio-resistance (radio-selected GPCs, rsGPCs). (**B**) rsGPCs and control GPCs were tested for colony formation capacities. The 500 cells/well were plated into a 24-well plate. On culture day 14, the cells were fixed with 2% formalin and counted under a light microscope. rsGPCs showed significantly enhanced colony formation capacities as compared to control GPCs. The results are the mean ± SD of three independent experiments with 4 replicates. The mean number of control GPC colonies was set to 100% and the data normalized to that mean. **** *p* < 0.0001. (**C**) Xenografts derived from GPCs or rsGPCs. Immunohistochemical staining for nestin. (**D**) rsGPCs and control GPCs were challenged to increasing doses of γ radiation as indicated and colony formation assay was performed. rsGPCs showed significantly enhanced colony formation capacities in contrast control GPCs at 2.5, 5.0, and 7.5 Gy of γ radiation. The means of both untreated control GPCs and rsGPCs were set to 100% and the treatment groups normalized accordingly. * *p* < 0.05; **** *p* < 0.0001.

**Figure 2 cancers-14-02728-f002:**
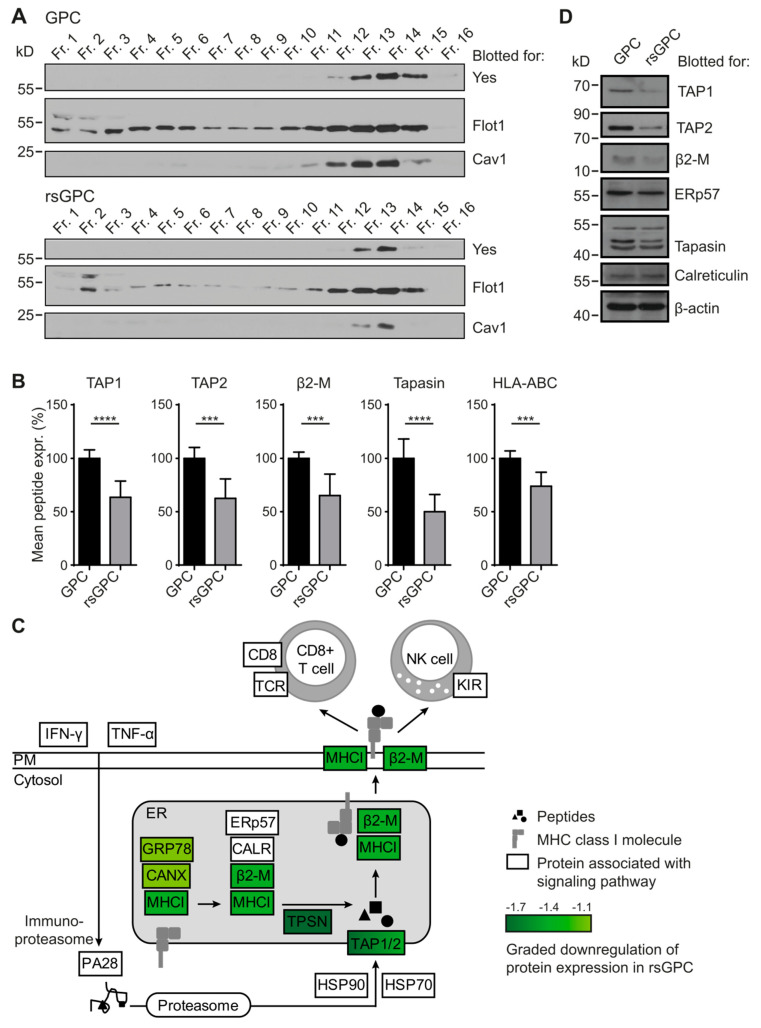
Downregulation of proteins involved in antigen processing and presentation. (**A**) To analyze possible changes in the composition of detergent-resistant membrane (DRM) proteins due to the radio-selection, GPCs and rsGPCs were lysed in 0.5% Brij98^®^ lysis buffer and fractionated on a discontinuous sucrose gradient. Fractions 12–15 contained DRM-associated proteins as shown by the expression of lipid raft markers such as Yes, Flot1 or Cav1 and were subsequently processed for mass spectrometric analysis. The results are representative of two independent experiments. (**B**) Analysis of DRM-associated proteins by mass spectrometry revealed that transporters associated with antigen processing (TAP1, TAP2), β2-Microglobulin (β2-M), Tapasin (Tpn), and MHC class I molecules (HLA-ABC) were among the most downregulated proteins in rsGPCs as compared to control GPCs. Values are illustrated as normalized mean ± SD of five technical replicates and two independent experiments. *** *p* < 0.001; **** *p* < 0.0001. (**C**) Various proteins associated with antigen processing and presentation were downregulated in rsGPCs as compared to control GPCs. The reduced fold change expression is depicted by a green graded color code. (**D**) Western blot analysis of proteins involved in antigen processing and presentation demonstrated that TAP1, TAP2, β2-Microglobulin, ERp57, Tapasin but not Calreticulin were downregulated in rsGPCs.

**Figure 3 cancers-14-02728-f003:**
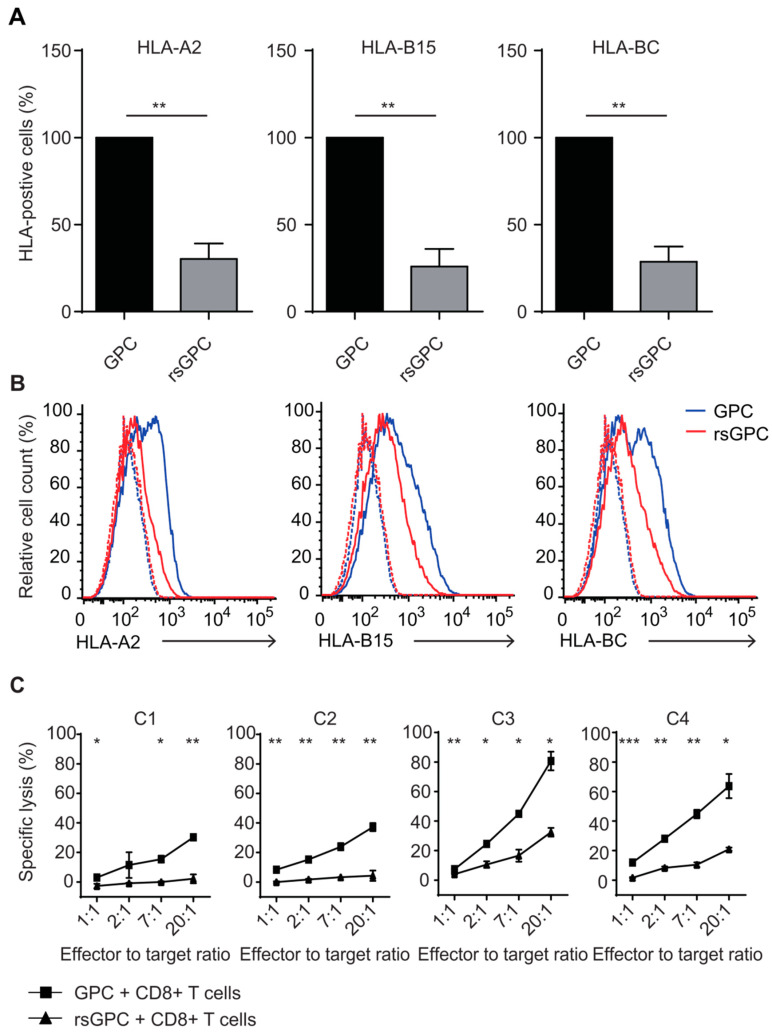
Fractionated radiation decreased MHC class I surface expression and recognition by CD8+ T cells. (**A**) Flow cytometric analysis of HLA molecules revealed a downregulation of HLA-class I, including the alleles HLA-A2, HLA-B15, and general HLA-BC alleles on the cellular surface of rsGPCs as compared to control GPCs. The results are depicted as % of HLA-positive cells ± SD of three independent experiments. (**B**) Flow cytometric analysis of HLA proteins on the surface of GPCs and rsGPCs revealed a downregulation of HLA-A2, HLA-B15, and pan-HLA-BC on rsGPCs. Depicted are histogram analyses of one out of three representative experiments. (**C**) HLA class I-partly matched CD8+ T-cell cultures were tested for their lysis potential of GPCs and rsGPCs in a ^51^Cr release assay. Depicted are four CD8+ cultures, stimulated with rsGPCs that were tested for lysis of GPCs (squares) and rsGPCs (triangles) with effector to target ratios from 20:1 to 1:1. Cultures C1 and C2 did not lyse rsGPCs but exhibited lysis activity against control GPCs. * *p* < 0.05; ** *p* < 0.01; *** *p* < 0.001.

## Data Availability

Data is contained within the article and Appendix A.

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
