# Peer review of "Gamma Irradiation Triggers Immune Escape in Glioma-Propagating Cells"

_cancers, 2022, doi:10.3390/cancers14112728_

Round 1
Reviewer 1 Report
The present manuscript "Gamma irradiation triggers immune escape in glioma-propagating cells" by Hoppmann et al. describes immunosuppressive effects in patient-derived glioma propagating cells upon fractionated radiation at clinically relevant but sublethal doses.
The manuscript is well written and the methodology seems sound.
Concern:
The relevance of the described findings would increase if the described effect could be recapitulated in vivo.
Author Response
Reviewer 1:
The present manuscript "Gamma irradiation triggers immune escape in glioma-propagating cells" by Hoppmann et al. describes immunosuppressive effects in patient-derived glioma propagating cells upon fractionated radiation at clinically relevant but sublethal doses.
The manuscript is well written and the methodology seems sound.
Concern:
The relevance of the described findings would increase if the described effect could be recapitulated in vivo.
Response:
We agree, that the in vitro findings should be proven in vivo, which would increase the value of the current results and is especially interesting regarding clinical translation. Nevertheless, stem cell-like glioma-propagating cells are the major drivers of tumor growth and recurrence and probably allow an in vivo transfer. We strongly suggest that our results are mainly regarded as important for the understanding of recurrent tumor cells because after radiotherapy most of these recurrences originate from the marginal zone of the irradiated field [1].
- Pei, J.; Park, I.-H.; Ryu, H.-H.; Li, S.-Y.; Li, C.-H.; Lim, S.-H.; Wen, M.; Jang, W.-Y.; Jung, S. Sublethal dose of irradiation enhances invasion of malignant glioma cells through p53-MMP 2 pathway in U87MG mouse brain tumor model. Radiation Oncology 2015, 10, 164, doi:10.1186/s13014-015-0475-8.

Reviewer 2 Report
The authors have preented a paper about "Gamma irradiation triggers immune escape in glioma propagating cells".
The topic is interesting and the manucript is overall well written however there are a few points which I would like the authors to address as follows:
1) Why was 2.5 Gy x 7 chosen as dose-fractionation? Was it dur the type of cell line used? didi the authors rely on previous literature? They should argue more clearly about it
2) In previous reports there is evidence supporting that use of low-dose radiotherapy in order to increase the response to systemic agents (see PMID 21993440 for instance)
3) Are the authors planning to use a higher number of fractions to see if the MHC behaviour changes?
4) It would be an addition to discuss why low-dose radiotherapy (below 1 Gy) and high-dose radiotherapy (single fraction) have proven to be immunogenic whereas and intermediate dose (17,5 Gy) has the effect to make the glioma propagating cells more resistant to radiotherapy
